# Can a Large Number of Transplanted Mesenchymal Stem Cells Have an Optimal Therapeutic Effect on Improving Ovarian Function?

**DOI:** 10.3390/ijms232416009

**Published:** 2022-12-16

**Authors:** Hyeri Park, Jin Seok, Jun Hyeong You, Dae Hyun Lee, Ja-Yun Lim, Gi Jin Kim

**Affiliations:** 1Department of Bioinspired Science, CHA University, Seongnam-si 13488, Gyeonggi-do, Republic of Korea; 2PLABiologicsCo., Ltd., Seongnam-si 13522, Gyeonggi-do, Republic of Korea; 3Department of Obstetrics and Gynecology, University of Chicago, 5841A. Maryland Ave, Chicago, IL 60637, USA; 4Department of Clinical Laboratory Science, Hyejeon College, 19 Daehak-1gil, Hongsung-eup, Hongsung-gun 32244, Chungnam-do, Republic of Korea

**Keywords:** mesenchymal stem cells, stem cells therapy, stem cells concentration, ovarian dysfunction, antioxidant

## Abstract

Mesenchymal stem cells (MSCs) are next-generation treatment in degenerative diseases. For the application of mesenchymal stem cell therapy to degenerative disease, transplantation conditions (e.g., optimized dose, delivery route and regenerating efficacy) should be considered. Recently, researchers have studied the mode of action of MSC in the treatment of ovarian degenerative disease. However, the evidence for the optimal number of cells for the developing stem cell therapeutics is insufficient. The objective of this study was to evaluate the efficacy in ovarian dysfunction, depends on cell dose. By intraovarian transplantation of low (1 × 10^5^) and high (5 × 10^5^) doses of placenta-derived mesenchymal stem cells (PD-MSCs) into thioacetamide (TAA)-injured rats, we compared the levels of apoptosis and oxidative stress that depend on different cell doses. Apoptosis and oxidative stress were significantly decreased in the transplanted (Tx) group compared to the non-transplanted (NTx) group in ovarian tissues from TAA-injured rats (* *p* < 0.05). In addition, we confirmed that follicular development was significantly increased in the Tx groups compared to the NTx group (* *p* < 0.05). However, there were no significant differences in the apoptosis, antioxidant or follicular development of injured ovarian tissues between the low and high doses PD-MSCs group. These findings provide new insights into the understanding and evidence obtained from clinical trials for stem cell therapy in reproductive systems.

## 1. Introduction

Mesenchymal stem cells (MSCs) represent a new form of treatment in degenerative disease and regenerative medicine. MSCs are capable of self-renewal and multilineage differentiation into various tissues (e.g., placenta, bone and adipose tissue) [1]. Additionally, they have a regenerative mechanism based on paracrine effects via signaling that changes the behavior of cells by cytokine secretion. Many scientists have reported the characteristics and therapeutic efficacy of stem cells in animal models with various degenerative diseases. Recently, MSCs were reported to improve follicular development and ovarian function by several modes of action (MoA) of MSCs in animal models with ovarian dysfunction [2]. MSCs activate signaling pathways by secreting cytokines and growth factors to improve ovarian function by reducing apoptosis and oxidative stress and increasing proliferation and vascular remodeling in the ovary. Kalhori et al. and their colleagues demonstrated that bone marrow MSCs (BM-MSCs) improved follicular development through their therapeutic efficacies including anti-inflammation, anti-apoptosis and angiogenesis in polycystic ovary syndrome (PCOS), known as ovarian metabolic disorder [3].

Oxidative stress is a representative cause of several degenerative diseases such as Alzheimer’s disease [4,5]. Imbalanced levels of antioxidants and reactive oxygen species (ROS), which areproduced by oxygen metabolism, lead to oxidative stress [6,7]. Their metabolic processes finally cause a rise in ROS levels, leading to oxidation of mitochondrial proteins, lipids and DNA [8]. Moreover, ROS have been demonstrated to be potentially critical for the induction and maintenance of the cell senescence process [9]. Excessive oxidative stress is one of the major causes in ovarian dysfunction. In the ovary, ROS inhibit steroidogenesis by uncoupling the luteinizing hormone receptor from adenylate. ROS also triggers apoptosis in the ovary as well as reduces the ovarian reserve by cytotoxicity to ovarian granulosa cells and oocytes. The ovaries become damaged due to this mechanism of oxidative stress [10]. Thus, many scientists have reported that oxidative stress is involved in the pathogenesis of PCOS. Oxidative stress promotes insulin resistance and arrests antral follicle development in PCOS [11,12]. Additionally, oxidative stress induces ovarian aging and POF through telomere shortening, apoptosis, inflammation, and ER stress [13,14]. To treat ovarian dysfunction, hormone replacement therapy (HRT) and metformin, which improve insulin sensitivity, are used. However, HRT shows temporal effects on ovarian reserve and increases the risks (e.g., osteoporosis, breast cancer). Metformin also regulates the menstrual cycle by improving ovarian metabolism but has the disadvantage of causing nausea and vomiting [15,16]. To overcome these limitations, many researchers have studied stem cell therapy for ovarian disease 2. In our previous reports, transplanted placenta-derived mesenchymal stem cells (PD-MSCs) improved follicular development by reducing oxidative stress in an ovariectomized rat model and a thioacetamide (TAA)-injured rat model [17,18].

Several clinical trials related to ovarian dysfunction (e.g., PCOS, POF and ovarian aging) based on the efficacy of these MSCs have been reported at http://www.clinicaltrials.gov (accessed on 22 June 2022). However, the most effective transplantation route, cell doses of MSCs and their homing ability after transplantation, have not yet been precisely elucidated. It is suggested that the transition of MSC-based therapy to the clinic can be accelerated if limitations on safety and therapeutic effectiveness are properly established [19]. Appropriate cell doses of MSCs per kilogram of body weight are being studied for clinical application [20]. Recently, many scientists attempted to apply the optimal cell number by transplanting MSCs in various degenerative diseases. For example, the number of transplanted mesenchymal stem cells is high for degenerative brain disease and cartilage disease. The sub-culture is necessary to secure a large number of cells to be transplanted by the blood circulation system. However, long-term subculture causes cell aging and chromosomal abnormalities, and decreases cell viability. Therefore, it is necessary to study the optimal cell number for each disease [21]. Kwon et al. and their colleagues studied the efficacy of low (1 × 10^6^ cells) and high (2 × 10^6^ cells) doses of human umbilical cord blood-derived mesenchymal stem cells (UCB-MSCs) in a rabbit model with a rotator cuff. They found no differences in regenerative effects between the high and low doses of MSCs [22]. According to Kim et al., a medium dose (5 × 10^4^ cells) of UCB-MSCs had significant therapeutic effects compared to a low (2.5 × 10^4^ cells) and a high (1 × 10^5^ cells) dose in mice with elastase-induced emphysema [23]. The optimal cell doses of MSCs are important for therapeutic efficacy. In this study, we aimed to determine whether the efficacy depends on doses of MSCs on an ovarian dysfunction rat model.

## 2. Results

### 2.1. Effect of Multi-Concentration PD-MSCs on Ovarian Function in the Ovaries of Ovarian Dysfunction Rats

We transplanted the low dose (1 × 10^5^ cells) and high dose (5 × 10^5^ cells) of PD-MSCs into the ovaries of TAA-injured rats (Figure 1A). To analyze the effect of several doses of PD-MSCs transplantation on an ovarian dysfunction rat model, we observed the ratio of ovary to body weight. The ratio of ovary to body weight is an index that can predict ovarian function and it was revealed that the ratio of TAA-injured ovaries was significantly decreased in the non-transplanted (NTx) group compared to the normal group. The low dose (1 × 10^5^ cells; 1x Tx) and high dose (5 × 10^5^ cells; 5x Tx) of PD-MSCs transplanted (Tx) groups were significantly increased compared to the NTx group (* *p* < 0.05; Figure 1B,C).

To confirm the engraftment of PD-MSCs, we analyzed the mRNA expression of human Alu (*hAlu*) in ovarian tissues. The mRNA expression of the *hAlu* was not detected in the NTx group. The mRNA expression of the *hAlu* was detected in the 1x Tx and 5x Tx groups (Figure 1D).

Ovarian function was regulated by the balance of endocrine hormones. Thus, we analyzed the sex hormone secretion, such as anti-Mullerian hormone (AMH) and estradiol (E2), in the individual serum of the TAA-injured rats. As a result, the levels of AMH and E2 in individual serum were significantly decreased in the NTx group compared to the normal group. The levels of AMH and E2 in individual serum samples was remarkably increased in 1x Tx and 5x Tx groups compared to the NTx group. Furthermore, the difference between 1x Tx and 5x Tx was not significant (* *p* < 0.05; Figure 1E,F). These results indicate that low dosages of MSCs sufficiently improve ovarian function.

### 2.2. Effect of a Low Dose of PD-MSCs on Apoptosis in the Ovaries of Rats with Ovarian Dysfunction

TAA, known to cause lipotoxicity, induces apoptosis [24]. To confirm the anti-apoptotic effect of transplanted PD-MSCs in ovarian tissue, we performed a TUNEL assay to detect dead cells in the ovaries of TAA-injured rats (Figure 2A). The apoptotic signal was dramatically increased in the NTx group compared to the normal group. Interestingly, the apoptotic signal was increased in granulosa cells (GCs) compared to theca cells (TCs) in the matured follicles of ovarian tissue. The apoptotic signal in theca cells was not significantly different between groups. In GCs, the Tx groups showed considerably decreased apoptotic signals compared to the NTx groups. Furthermore, the 1x Tx group had a significantly decreased apoptotic signal compared to the 5x Tx group (* *p* < 0.05; Figure 2B,C).

To confirm cytotoxicity, we analyzed cell death in serum from individual TAA-injured rats using a lactate dehydrogenase (*LDH*) assay. The cytotoxicity level was significantly increased in the NTx group compared to the normal group, and the Tx groups had markedly decreased cytotoxicity levels compared to the NTx group (* *p* < 0.05; Figure 2D). These results suggest the possibility that a low dose of MSCs has an anti-apoptosis effect similar to that of a high dose of MSCs.

### 2.3. Effect of a Low Dose of PD-MSCs on Oxidative Stress in the Ovaries of Rats with Ovarian Dysfunction

Oxidative stress is known to increase in damaged cells [25]. To confirm the ROS level, we stained ovarian tissues with MitoSOX, which is a mitochondrial superoxide indicator (Figure 3A). To quantify the MitoSOX-positive signal, we analyzed the ratio of MitoSOX to MitoTracker expression in the mature follicles of individual ovarian tissues. The ratio of MitoSOX to MitoTracker expression was markedly increased in the NTx group compared to the normal group and significantly decreased in the Tx groups compared to the NTx group (* *p* < 0.05;Figure 3B). 

To demonstrate oxidative stress through oxidase factors, we analyzed the expression of mRNAs related to oxidative stress in the ovarian tissues of individual TAA-injured rats. The mRNA expression of NADPH oxidase (*NOX4*), which produces ROS, was significantly increased in the NTx group compared to the normal group. However, the mRNA expression of *NOX4* was considerably decreased in the Tx groups compared to the NTx group (* *p* < 0.05; Figure 3C). The mRNA expression of beta-subunit prolyl 4-hydroxylase (*P4hb*), which acts an oxidoreductase, was markedly increased in the NTx group compared to the normal group, and the mRNA expression of *P4hb* was significantly decreased in the Tx groups compared to the NTx group (* *p* < 0.05; Figure 3D). 

To confirm the antioxidant effect through the antioxidant factors of PD-MSCs, we demonstrated the mRNA expression related to antioxidants in individual ovarian tissues of TAA-injured rats. The mRNA expression of heme oxygenase-1 (*HO-1*), which is an enzyme responsible for the breakdown of heme, was significantly decreased in the NTx group compared to the normal group. The mRNA expression of *HO-1* was significantly increased in the Tx groups compared to the NTx group (* *p* < 0.05; Figure 3E). The mRNA expression of superoxide dismutase type 1 (*SOD1*), which protects the cell from ROS, was decreased in the NTx group compared to the normal group, and the mRNA expression of *SOD1* was increased in the Tx groups compared to the NTx group (Figure 3F). Additionally, the mRNA expression of superoxide dismutase type 2 (*SOD2*), which converts a damaging agent into hydrogen peroxide, was considerably decreased in the NTx group compared to the normal group, and the mRNA expression of *SOD2* was markedly increased in the Tx groups compared to the NTx group (* *p* < 0.05; Figure 3G). Furthermore, the mRNA expression of *catalase*, an enzyme that protects the cell from ROS, was significantly decreased in the NTx group compared to the normal group, and the mRNA expression of *catalase* was notably increased in the Tx groups compared to the NTx group (* *p* < 0.05; Figure 3H). These results suggest that a low dose of MSCs has antioxidant effects similar to those of a high dose of MSCs.

### 2.4. Effect of a Low Dose of PD-MSCs on Follicular Development in the Ovaries of Rats with Ovarian Dysfunction

To evaluate whether follicular development was improved through antioxidant effects, we analyzed the genes related to follicular development in the individual ovarian tissues of TAA-injured rats. The mRNA expression of KIT ligand (*Kitlg*), which promotes the initiation and progression of primordial follicles as a stem cell factor (*SCF*) receptor, was significantly decreased in the NTx group compared to the normal group. After transplanting of MSCs 1x and 5x, the mRNA expression of *Kitlg* was significantly increased in the Tx groups compared to the NTx group (* *p* < 0.05; Figure 4A). Additionally, the mRNA expression of epidermal growth factor receptor (*EGFR*), which is involved in follicular maturation, was significantly decreased in the NTx group compared to the normal group, and the mRNA expression of *EGFR* significantly increased in the Tx groups compared to the NTx group (* *p* < 0.05; Figure 4B). These results show that a low dose of MSCs induces follicular maturation similar to a high dose of MSCs.

To confirm the effect on follicular development, we examined serially sectioned ovarian tissues from each group by H&E staining (Figure 4C). Then, using the 3D HISTECH program, the number of follicles present at each stage in the ovarian tissue was counted, and follicles were counted by three different people for confirmation (Table 1). Mature follicles (e.g., secondary and antral follicles) were significantly decreased in the NTx group compared to the normal group. In addition, atresia follicles were increased in the NTx group compared to the normal group. After PD-MSC transplantation, each stage of follicles (e.g., primordial, primary, secondary, antral and atretic) was protected. The mature follicles (e.g., secondary and antral follicles) were increased in the Tx groups compared to the NTx group, and atresia follicles were decreased in the Tx groups compared to the NTx group. Interestingly, antral follicles were significantly increased in 5x Tx group compared to the NTx group. Secondary follicles, which are follicles in the maturation stage, were also increased in the 1x Tx group compared to the 5x Tx group (* *p* < 0.05; Table 1, Figure 4D). These results indicate that a low dose of MSCs improves follicular development and protects each follicle in a manner similar to that of a high dose of MSCs.

## 3. Discussion

MSCs therapy is spotlighted as a next-generation therapeutic to improve function in ovarian-related diseases (e.g., POF, ovarian aging). Recently, many scientists reported the various MSCs therapies on ovarian dysfunction [26]. Jalalie et al. demonstrated that the UC-MSCs improve morphometric of follicles in a POF-induced mouse model. Mohamed et al., demonstrated that BM-MSCs reactivate the follicular development through improved hormone secretion [27,28]. Despite the finding that MSCs improve ovarian function, MSCs have a weaker MoA in improving ovarian function. To overcome these issues, many researchers have studied general phenotypes and paracrine mechanisms in degenerative diseases [29]. Thus, we analyzed the oxidative stress, which is a common and representative mechanism in ovarian disease. In our previous report, we confirmed that intravenously transplanted PD-MSCs (2 × 10^6^ cells/ea) improved ovarian function in a TAA-injured rat model by activating antioxidant factors [18]. Although it has been confirmed that PD-MSCs are effective in improving ovarian function through antioxidant effects, studies on the number of cells and transplantation routes are required for their development as clinically applicable therapeutics.

Efforts to apply MSCs-based therapy not only to preclinical animal models but also to clinical practice are continuing [30]. However, clinical studies on MSC-based therapy for ovarian dysfunction (e.g., POF, PCOS) are lacking. Although there is no basis for optimal cell doses of MSC therapy, appropriate doses of MSC therapy in animal models related to POF have been reported. The doses of MSCs, in the thousands, millions, or billions based on the recipient’s body weight, were regarded as necessary for a therapeutic effect because of the initial perception of cells as a drug [31]. Therefore, the capacity setting of the cell is important. Liu and colleagues used a low dose (5 × 10^6^ cells/kg), middle dose (7.5 × 10^6^ cells/kg) and high dose (1 × 10^7^ cells/kg) of human amnion-derived MSCs and implanted them into the tail vein of POF mice. Among the three concentrations, hAMSCs at the middle dose (7.5 × 10^6^ cells/kg) increased AMH and estrogen and significantly decreased FSH compared to the POF group [32]. In our report, we transplanted the lower cell dose compared to Liu et al. We had a limit in the volume that couldn’t be transplanted with excessive cells for intraovarian transplantation, which is safer than intravenous transplantation. Although we transplanted much fewer PD-MSCs than Liu’s study, our result showed that the AMH level was significantly increased compared to the NTx group and improved to a level similar to that of normal. Furthermore, the 1x Tx of PD-MSCs showed effects similar to those of 5x Tx in terms of sex hormones, follicular development and antioxidants. These results refute the notion that transplantation of high-dose MSCs is superior to low-dose MSCs transplantation and suggest the possibility of low-dose transplantation.

In addition to setting the cell capacity, there are other things to consider, such as the number of transplants and the route of transplantation. According to Lv and colleagues, both the group transplanted with UC-MSCs once and the group transplanted three times had improved granulosa proliferation and AMH levels compared to the POF group [33]. To analyze the improvement of ovarian function by transplanted UC-MSCs, Song et al. performed intravenous transplantation (1 × 10^5^ cells/ea), whereas Mohamed et al. performed ovarian transplantation (5 × 10^4^ cells/per ovary) [34,35]. Both transplantation routes have been shown to improve ovarian function. In addition, in our previous report, we analyzed intravenous transplantation of PD-MSCs with low (1 × 10^5^ cells/ea), middle (5 × 10^5^ cells/ea) and high (2 × 10^6^ cells/ea) doses. The high dose of PD-MSCs significantly increased the hormones secretion (e.g., AMH, estrogen) compared to the low-dose, middle–dose, and the NTx groups, but no significant differences were observed in follicular development and antioxidant effects. Rather, LDH levels were significantly increased in high doses of PD-MSCs compared to low and middle doses of PD-MSCs (data not shown). However, intravenously administered MSCs carry the risk of first accumulating in the lung vasculature [36]. Intravenously administered MSCs have the limitation of a lower rate of engraftment by delivery to various organs through the blood flow. Local transplantation is an advantage that allows cell delivery into the targeted tissues [37]. From a homing perspective, we thought the adequate MSC delivery route oflocal transplantation has a stronger advantage than intravenous delivery for small tissues such as ovaries. Therefore, we suggest the route of direct transplantation into the organ for safety purposes.

Finally, we confirmed that the low dose of MSCs had similar or superior therapeutic efficacy to the high dose of MSCs in the ovarian tissues of rats with ovarian dysfunction through the antioxidant effect. This will provide supporting evidence for an optimal therapeutic effect in improving ovarian function.

## 4. Materials and Methods

### 4.1. Cell Culture

Human placental samples were approved by the Institutional Review Board of CHA Gangnam Medical Center, Republic of Korea (IRB-07-18). According to our previous reports, PD-MSCs were isolated from the chorionic plate of the placenta [38]. PD-MSCs were cultured in alpha-minimum essential medium (α-MEM; HyClone, UT, USA) containing 10% fetal bovine serum (FBS; Gibco-BRL, OK, USA), 1% penicillin/streptomycin (P/S; Gibco-BRL), 25 ng/mL FGF-4 (Peprotech, NJ, USA), and 1 μg/mL heparin (Sigma–Aldrich, MS, USA) at 37 °C in a humidified atmosphere of 5% CO_2_. After harvest, a low dose (1 × 10^5^ cells/ea; 1x Tx) of PD-MSCs and a high dose (5 × 10^5^ cells/ea; 5x Tx) of PD-MSCs were injected into rats via the intraovarian route in the transplanted animal group.

### 4.2. Animal Construction

All animal experiments were approved by the Institutional Animal Care and Use Committee (IACUC 210005) of the CHA Laboratory Animal Research Center in the Republic of Korea. Female 7-week-old female Sprague–Dawley rats (Koatech, Pyeongtaek, Republic of Korea) were maintained in an air-conditioned animal facility at 21 °C. A12 h/12 h light-dark cycle was employed. Groups of two rats were housed in plastic cages with corn-cob bedding and provided ad libitum access to standard commercial food and tap water. The ovarian disorder model was established by intraperitoneal injection of 300 mg/kg thioacetamide (TAA; Sigma–Aldrich), a metabolic disorder inducer, twice a week for 12 weeks. After 8 weeks of injections, a low dose (1 × 10^5^ cells/ea) and a high dose (5 × 10^5^ cells/ea) of PD-MSCs were transplanted via the intraovarian route. PD-MSCs were distributed into each ovary. PD-MSCs of 1 × 10^5^ cells were transplanted at 5 × 10^4^ cells per ovary, and PD-MSCs of 5 × 10^5^ cells were transplanted at 2.5 × 10^5^ cells per ovary. To reduce ovarian damage, PD-MSCs were injected in the direction of the ovaries through the oviduct rather than directly into the ovaries. Rats were anesthetized with avertin (2,2,2-tribromoethanol; Sigma–Aldrich) before transplantation. Four weeks after transplantation, all rats were sacrificed, and their ovarian tissues and blood were harvested. We sacrificed 5 rats of the normal group, 3 rats of the NTx group, 6 rats of the 1x Tx group and 7 rats of the 5x Tx group. 

### 4.3. RNA Isolation and Quantitative Real-Time Polymerase Chain Reaction (qRT-PCR)

The ovarian tissues were homogenized with liquid nitrogen. Total RNA was isolated from individual rat ovarian tissues using TRIzol reagent (Ambion, Thermo Fisher Scientific, Waltham, MA, USA) according to the manufacturer’s protocol. The total RNA concentration was measured by means of a Nanodrop spectrophotometer (Thermo Fisher Scientific) and reverse transcribed into cDNA using Superscripts III reverse transcriptase (Invitrogen). The PCR conditions for the synthesis of cDNA were as follows: 5 min at 65 °C, 1 min at 4 °C, 60 min at 50 °C, and 15 min at 72 °C. The cDNA was used for qRT–PCR analysis with SYBR Ex Taq (Roche, Basel, Switzerland). The cDNA was amplified by PCR under the following conditions: 5 s at 95 °C, 40 cycles of 95 °C for 5 s and 60 °C for 30 s. The sequences of the qRT–PCR primers are listed in Table 2. r*GAPDH* was used as an internal control for normalization, and each sample was analyzed in triplicate.

### 4.4. ELISA

Individual blood samples were collected from the aortas of rats in all groups that had been fasted one day before. Serum samples were separated from whole blood by using a blood collection tube (Vacutainer; BD Biosciences, NJ, USA). All blood serum was stored at −80 °C, and anti-Müllerian hormone (AMH) (Elabscience Biotechnology, TX, USA), estrogen (BioVision, Milpitas, MA, USA) and lactate dehydrogenase (LDH) (Abcam, Waltham, MA, USA) activity in serum was analyzed by ELISA kits following the manufacturer’s instructions.

### 4.5. H&E Staining

Ovarian tissues were fixed with 10% neutral buffered formalin (BBC, Washington, IL, USA) and embedded in paraffin. The paraffin block was serially cut into 4-μm ovary sections and these tissues were deparaffinized in a 60 °C dry oven and with xylene and ethanol. The deparaffinized tissues were then washed under tap water. The specimens were stained with Harris hematoxylin (Leica Biosystems, Wetzlar, Germany) for 7 min, and the specimens were dipped in 0.1% HCl for 2 s and counterstained with alcoholic eosin Y solution (Sigma–Aldrich). The stained slides were scanned for whole ovaries by 3D HISTECH (The Digital Pathology Company, Budapest, Hungary). The follicles were counted at 100-μm intervals in serially sectioned slides, and the total follicles were defined as primordial, primary, secondary, antral and atretic follicles, according to previous reports [39]. To analyze the number of follicles in ovarian tissue, the number of each follicle was counted.

### 4.6. TUNEL Assay

The paraffin block containing the ovary was cut into 4-μm sections, and these tissues were deparaffinized in a 60 °C dry oven by xylene and ethanol. Deparaffinized tissues were then washed with 1X Tris-buffered saline-Tween 20 (TBS-T) and stained using a TUNEL assay kit (Abcam, Cambridge, England) following the manufacturer’s instructions.

### 4.7. MitoTracker & MitoSOX Staining

Frozen blocks of ovary tissue were cut into 7-μm sections and fixed with methanol for 20 min. The fixed ovarian tissues were then washed with 1X phosphate-buffered saline (PBS) and incubated with Hank’s balanced salt solution (HBSS) containing 1.5 μMMitoSOX (Invitrogen) and 50 nMMitoTracker (Invitrogen) for 40 min at 37 °C in a dry oven. The specimens were washed with 1X PBS. The slides containing ovarian tissues were observed with a fluorescence microscope. All parts of each slide were observed, and representative images were captured and analyzed by the ImageJ program (Wayne Rasband, Bethesda, MA, USA).

### 4.8. Statistical

The experiments were conducted in duplicate or triplicate. The results are presented as the mean ± standard error. GraphPad Prism 5.0 (GraphPad Software, Inc., CA, USA) was used to conduct statistical analysis using one-way ANOVA followed by Tukey’s multiple comparisons test, with a *p* value less than 0.05 indicating statistical significance.

## 5. Conclusions

The intraovarian transplantation of low and high dose PD-MSCs improved the follicular development in the ovaries of rats with ovarian dysfunction via antioxidant and antiapoptotic effects. Interestingly, low doses of PD-MSCs showed antioxidant and antiapoptotic effects at similar levels to high doses of PD-MSCs.

In conclusion, our findings indicate that a low dose of MSCs has similar or superior efficacy to high doses, and this refutes the prejudice against low doses of MSCs. Furthermore, it provides an opportunity for us to remind ourselves about the considerations for safe MSC therapy, including the number of cells and transplantation routes. Therefore, these findings offer new avenues into stem cell therapy for reproductive medicine.

## Figures and Tables

**Figure 1 ijms-23-16009-f001:**
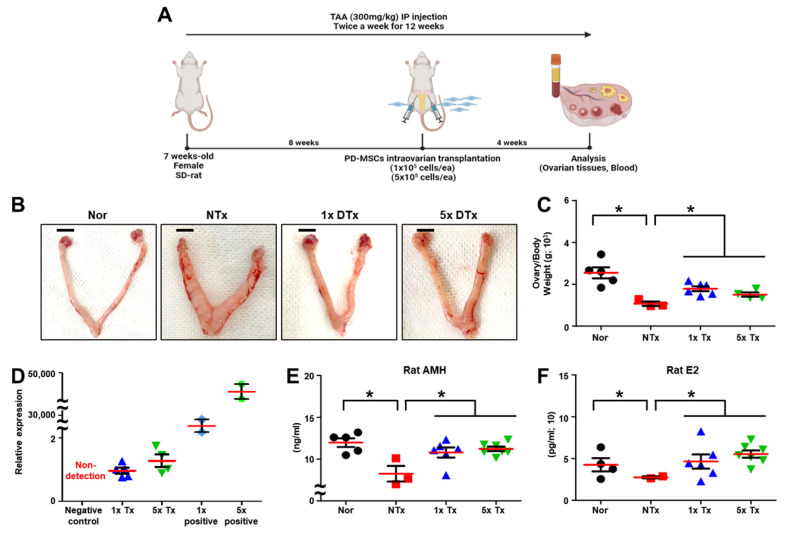
Effect of a low dose of PD-MSCs on ovarian function in rats with ovarian dysfunction. (**A**) Scheme of TAA-injured rat model construction. (**B**) Morphological analysis in ovarian tissues of TAA-injured rats after sacrifice. (**C**) The ratio of ovary weight to body weight was analyzed after sacrifice. (**D**) The mRNA expression of *hAlu* was analyzed by qRT-PCR. (**E**) The concentrations of AMH and (**F**) E2 in individual serum samples were analyzed by ELISA. The data are representative of three independent experiments and expressed as the mean ± S.D. Statistical significance was determined by using one-way ANOVA, * *p* < 0.05.

**Figure 2 ijms-23-16009-f002:**
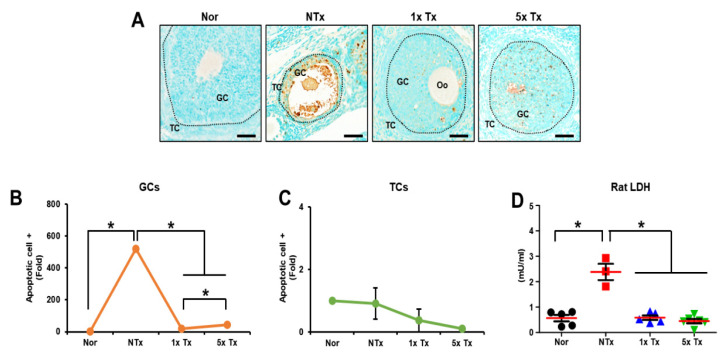
Effect of a low dose of PD-MSCs on apoptosis in the ovaries of rats with ovarian dysfunction. (**A**) The apoptotic cells of follicles were stained with a TUNEL assay kit. Scale bar: 100 μm, Magnification: 20×. (**B**) Apoptosis in granulosa cells (GCs) and (**C**) theca cells (TCs) of mature follicles (e.g., secondary, antral follicles) was quantified by means of the 3D HISTECT program. (**D**) The concentrations of LDH in individual serum samples were analyzed by ELISA. GC: granulosa cell, TC: theca cell, Oo: oocyte. The data are representative of three independent experiments and expressed as the mean ± S.D. Statistical significance was determined by using one-way ANOVA, * *p* < 0.05.

**Figure 3 ijms-23-16009-f003:**
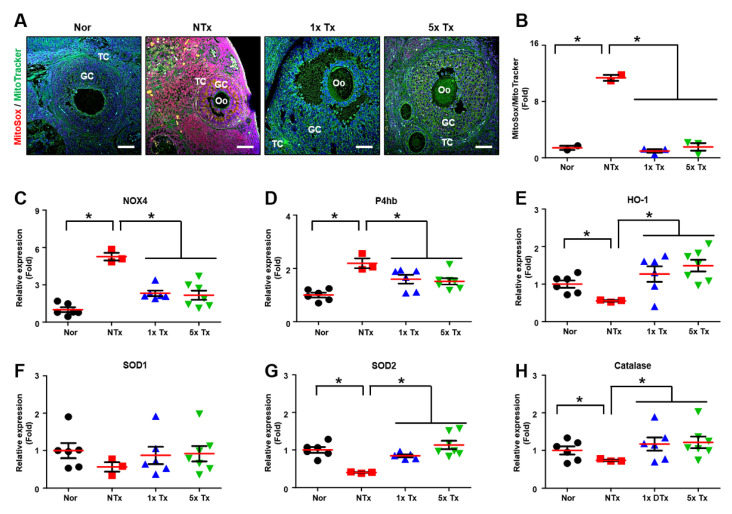
Effect of a low dose of PD-MSCs on oxidative stress in the ovaries of rats with ovarian dysfunction. (**A**) The ovarian tissues of TAA-injured rats were stained with MitoSOX and mitoTracker. Scale bar: 20 μm, Magnification: 20×. (**B**) The intensity of MitoSOX/MitoTracker was quantified by means of the Image J program. (**C**) The mRNA expression levels of *NOX4*, (**D**) *P4hb*, (**E**) *HO-1*, (**F**) *SOD1*, (**G**) *SOD2* and (**H**) *catalase* were analyzed by qRT-PCR. GC: granulosa cell, TC: theca cell, Oo: oocyte. The data are representative of three independent experiments and expressed as the mean ± S.D. Statistical significance was determined by using one-way ANOVA, * *p* < 0.05.

**Figure 4 ijms-23-16009-f004:**
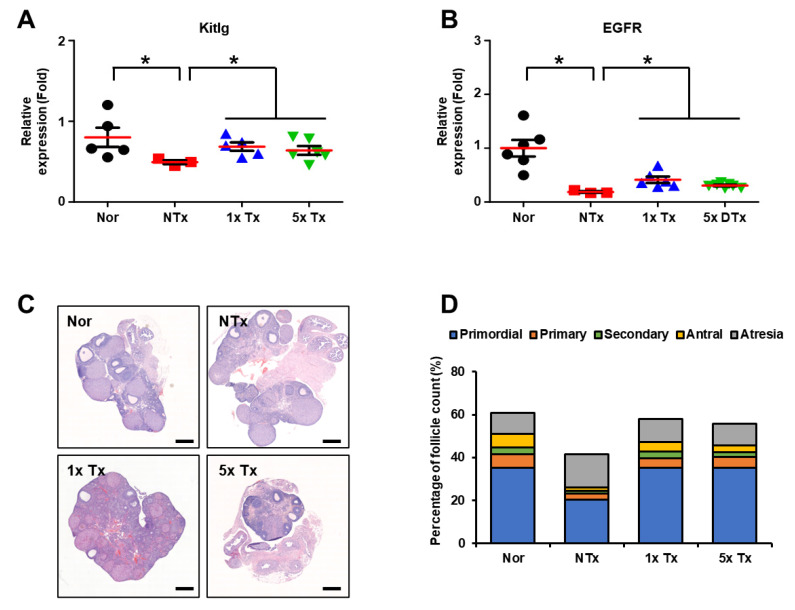
Effect of a low dose of PD-MSCs on follicular development in the ovaries of rats with ovarian dysfunction. (**A**) The mRNA expression levels of Kitlg and (**B**) EGFR were analyzed by qRT-PCR. (**C**) Histological analysis of follicular development in the ovary was analyzed by H&E staining. (**D**) The follicle count according to follicular development was analyzed by the 3D HISTECH program. Scale bar: 500 μm; magnification: 2×. The data are representative of three independent experiments and expressed as the mean ± S.D. Statistical significance was determined by using one-way ANOVA, * *p* < 0.05.

**Table 1 ijms-23-16009-t001:** Comparison of follicle counts after transplantation.

	Primordial	Primary	Secondary	Antral	Atresia
Normal (*n* = 3)	35.38 ± 2.01	6.07 ± 1.49	3.30 ± 0.81	6.36 ± 1.09	9.67 ± 2.28
NTx (*n* = 2)	20.29 ± 6.02 *	2.90 ± 0.36	1.35 ± 0.48 *	1.55 ± 0.41 *	15.35 ± 0.42
1x Tx (*n* = 3)	35.33 ± 4.11 **	4.28 ± 0.67	3.11 ± 0.23 ^#^	4.59 ± 0.84	10.80 ± 3.80
5x Tx (*n* = 4)	35.12 ± 6.05 **	5.04 ± 0.93	2.25 ± 0.52	3.24 ± 0.15 **	10.26 ± 4.46

*, NTx vs. Nor (*p* < 0.05). **, Tx vs. NTx (*p* < 0.05). ^#^, 1x Tx vs. 5x Tx (*p* < 0.05).

**Table 2 ijms-23-16009-t002:** Primer sequences using quantitative real time polymerase chain reaction.

Gene	Primer	Annealing Temperature (°C)	NM Number
*hAlu*	F: 5′-GGA GGC TGA GGC AGG AGA A-3′	60	NM_002715
	R: 5′-CGG AGT CTC GCT CTG TCG CCC A-3′		
*Catalase*	F: 5′-TCA GAG GAA AGC GGT CAA GA-3′	58	NM_012520.2
	R: 5′-CCC GTG CTT TAC AGG TTA GC-3′		
*HO-1*	F: 5′-TGC ACA TCC GTG CAG AGA AT-3′	59	NM_012580.2
	R: 5′-CTG GGT TCT GCT TGT TTC GC-3′		
*SOD2*	F: 5′-AGC TGC ACC ACA GCA AGC AC-3′	62	NM_017051.2
	R: 5′-TCC ACC CTT AGG GCT CA-3′		
*NOX4*	F: 5′-AGG TGT CTG CAT GGT GGT G-3′	58	NM_053524.1
	R: 5′-GAG GGT GAG TGT CTA AAT TGG T-3′		
*P4hb*	F: 5′-AGC TGC CTT TGG TCA TCG AG-3′	59	NM_012998.2
	R: 5′-AGT ATG CGC TGG TTG TCA GT-3′		
*EGFR*	F: 5′-AGA TTG CAA AGG GCA TGA ACT AC-3′	59	NM_001393707.1
	R: 5′-ACA TTC cTG CT GCC AAG TC-3′		
*Kitlg*	F: 5′-CAG CCA GTT CCC TTA GGA ATG A-3′	58	NM_021843.4
	R: 5′-AGC AAA GCC AAT TAC AAG CGA-3′		
*GAPDH*	F: 5′-TCC CTC AAG ATT GTC AGC AA-3′	55	NM_017008.4
	R: 5′-AGA TCC ACA ACG GAT ACA TT-3′		

## Data Availability

The data presented in this study are available on request from the corresponding author.

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
