# Peer review of "Can a Large Number of Transplanted Mesenchymal Stem Cells Have an Optimal Therapeutic Effect on Improving Ovarian Function?"

_ijms, 2022, doi:10.3390/ijms232416009_

Round 1

Reviewer 1 Report

The authors have tested the importance of cell dose of MSC administered to rats to rescue ovarian health post thioacetamide injury. The studies were straight forward, following approaches previously used by this group. Doses and route of administration were chosen based upon prior published work in the field and on that of the authors. Multiple aspects of overall ovarian health and follicle development were conducted. Appropriate statistical approaches were used throughout and formed the basis on assessing significance of the variables tested. 

My only question is whether the authors can more carefully place their work in context of some of the published work in the field that is based on cell count/kg rather than on absolute cell number. Is any data available from the author's prior work to indicate how many IV transplanted MSC actually engrafted in the ovaries compared to the doses of cells directly administered in this paper?

Author Response

Dear Reviewer,

Thank you for your kind supporting.

We were really encouraged by the reviewers’ positive comments and constructive suggestions. I am happy to report that we have successfully addressed all issues and subsequent revision of our manuscript, as detailed in the following response page.
As Reviewer’s commented, we corrected it clearly stating with each comment and changes are highlighted in red in the revised manuscript.

Very sincerely yours,

Gi Jin Kim, Ph.D.

Associate Professor
Dept. of Biomedical Science
CHA University

Reviewer 2 Report

Only 2 rats were used per group. is this correct? How do you justify the results with just 2 animals per group? How do you have 5-6 data points in Nor, 1X TX, 5X TX and only 3 data points in all the graphs if you had only 2 mice per group? 

Line 100: Typing error : voarian

Can a cartoon/figure be made to show the animal construction?

What is the rationale for looking at the ovary/body weight after the transplantation of PD-MSC and not just the ovary? 

What is the correct dose of PD-MSC transplanted? The abstract and result section mention Low dose (1x105 cells; 1x Tx) and high dose (5x105 cells; 5x Tx) of PD-MSCs transplanted while the methods mention something different - low dose (5x104 cells/ovary) and a high dose (2.5x105 cells/ovary) of PD-MSCs were transplanted via the intraovarian route.  PLEASE CORRECT THIS ERROR

What do you mean by the MSC were transplanted? Is it same as injecting them intrabursal in the ovaries? if not how is it different?

How effective was the PD-MSC transplantation in the ovaries? Was a dye or tracker used to detect if the implants were made correctly? if not are there other means that were used to detect the presence of MSC in ovaries after 4 weeks if PD-MSC injections other than mRNA expression? Do you suspect the MSC to move out of the ovaries and thus have less effect?

Please mention the difference between the Nor mice and NTx? Are the NTx injected with TAA and Nor mice are not?

Figure 2 B: can the graph be split to see the numbers 0-200 in order to appreciate the significant difference between 1 and 5X TX as done in 1C. 

Figure 2: Have you looked at the number of live cells in your groups and do they correlated with the dead cell graph?

https://www.ncbi.nlm.nih.gov/books/NBK540958/ 

Figure 3A - Need more data sets. Seems like only 2 sections were analyzed and stopped after significance was reached. Please perform a biochemical assay for ROS. I am sure there are several kits available.  

Was the gene expression related to antioxidants evaluated by Western blot?

Figure 4 A and B have typing error on the graph title as compared to the result section. 4A is Kitlg expression in the result but the graph is 4B in the figure and visa-versa for EGFR. Please correct. 

Was the gene expression by Western blot?

Do you believe higher MSC dose is required if injected IP or IV versus implantation at a particular site? Also, do MSC derived from different sites like PD, UC, BM etc have the same effect on a disease? comment. why not derive MSC from different areas and then treat a disease in mouse model for the efficacy? 

Author Response

(The authors gave the same response as above.)

Round 2

Reviewer 2 Report

Thank you for revising the manuscript. Most of my questions/suggestions were answered. I found some minor corrects. They are below.:

Line 336: We sacrificed 5 rats of normal group, 3 rats of normal group, 6 rats of 1x Tx group and 7 rats of 5x Tx group. - change the bold to NTx group.

As you have mentioned that you have confirmed using the dye the injection of PD-MSC's. Can this be incorporated in this manuscript? If not, it's fine. 

The manuscript will be accepted after the change in line 336.

Author Response

Reviewer #2:

Comments and suggestions for authors:

Thank you for revising the manuscript. Most of my questions/suggestions were answered. I found some minor corrects. They are below.:

Author’s reply:

We greatly appreciate the reviewer’s positive statement that “Can a large number of transplanted mesenchymal stem cells have an optimal therapeutic effect on improving ovarian function?”.  We revised the typo through your comments.

Point #1: Line 336: We sacrificed 5 rats of normal group, 3 rats of normal group, 6 rats of 1x Tx group and 7 rats of 5x Tx group. - change the bold to NTx group.

  • Author’s response:

Thank you for your critical comments. The tying error was corrected in the revised manuscript.

Point #2: As you have mentioned that you have confirmed using the dye the injection of PD-MSC's. Can this be incorporated in this manuscript? If not, it's fine. 

  • Author’s response:

We greatly appreciate the reviewer bringing up this important point. Although our preliminary experiment with the dye was only a check and preliminary experiment for local injection before stem cell transplantation, we used PKH67 labeling and human Alu sequence analyses to confirm engraftment of PD-MSCs in the ovary in the present study. Especially, the result for human Alu (hAlu) sequence expression was included in the Figure 1D of the manuscript because it was thought that it could be accurately show engraftment of PD-MSCs into the ovary, as shown in Figure 1D.
